ecology, evolution, plant science

bryophyte, *Ceratodon purpureus*, fitness, fertilization, mating system, microarthropods

**Author for correspondence:**
Erin E. Shortlidge
e-mail: eshortlidge@pdx.edu

# Microarthropod contributions to fitness variation in the common moss *Ceratodon purpureus*

Erin E. Shortlidge[1], Sarah B. Carey[2], Adam C. Payton[2], Stuart F. McDaniel[2], Todd N. Rosenstiel[1] and Sarah M. Eppley[1]

[1]Department of Biology, Portland State University, PO Box 751, Portland, OR 97202-0751, USA
[2]Department of Biology, University of Florida, PO Box 118525, Gainesville, FL 32611-8525, USA

 EES, 0000-0001-8753-1178; SFM, 0000-0002-5435-7377

The evolution of sustained plant–animal interactions depends critically upon genetic variation in the fitness benefits from the interaction. Genetic analyses of such interactions are limited to a few model systems, in part because genetic variation may be absent or the interacting species may be experimentally intractable. Here, we examine the role of sperm-dispersing microarthropods in shaping reproduction and genetic variation in mosses. We established experimental mesocosms with known moss genotypes and inferred the parents of progeny from mesocosms with and without microarthropods, using a pooled sequencing approach. Moss reproductive rates increased five-fold in the presence of microarthropods, relative to control mesocosms. Furthermore, the presence of microarthropods increased the total number of reproducing moss genotypes, and changed the rank-order of fitness of male and female moss genotypes. Interestingly, the genotypes that reproduced most frequently did not produce sporophytes with the most spores, highlighting the challenge of defining fitness in mosses. These results demonstrate that microarthropods provide a fitness benefit for mosses, and highlight the potential for biotic dispersal agents to alter fitness among moss genotypes.

## 1. Introduction

Gene flow shapes genetic variation within and among populations, thereby influencing long-term patterns of adaptation and speciation [1–3]. Sessile organisms employ a variety of strategies to promote outcrossing and gene flow. Many angiosperms [1,4–6] rely on biotic agents to disperse pollen from one plant to another [e.g. 7–12], which assures seed production, and can promote outcrossing while reducing the number of gametes lost in interspecies mating. In return, the pollinators gain resources themselves [13–15]. Animals may also disperse gametes in other plant groups, such as cycads and bryophytes. Although these plants release water-dispersed motile sperm [16–20], naturally occurring microarthropods [21,22] such as Oribatid mites and common springtails, (Collembolan species *Folsomia candida* and *Sinella curviseta*), can enhance sexual reproduction (i.e. sporophyte formation) in laboratory moss cultures [23,24]. Remarkably, springtails also choose female mosses over male mosses in olfactory choice tests [24]. These observations suggest that mosses and microarthropods participate in scent-mediated fertilization syndrome, much like angiosperms and their pollinators. Yet, we know little about the influence of this interaction on the fitness variation of mosses under natural conditions [25,26].

Mosses provide an experimentally tractable system in which to explore the fitness consequences of interactions between plants and gamete-dispersing animals. Many mosses have separate sexes, and therefore are obligate outcrossers. Mosses additionally can be clonally propagated, meaning that experimental

arrays containing the exact same genotypes can be exposed to multiple treatments. Here, we constructed replicated outdoor experimental mesocosms composed of known male and female genotypes of the moss *Ceratodon purpureus*, and we manipulated the presence of field-collected microarthropods [27–31]. We found that microarthropods dramatically increased reproduction, increased the number of moss genotypes that reproduced, and differentially influenced the fitness of some moss genotypes. These findings suggest that mosses may form a potentially ancient commensal relationship with sperm-dispersing microarthropods. These data also highlight the potential for microarthropods in maintaining genetic variation for fitness among terrestrial bryophytes.

## 2. Material and methods

### (a) Study populations
We collected *C. purpureus* gametophytes from three populations in and near Portland, OR, USA. We air-dried samples from each population and isolated single gametophytes for further study. Using a dissecting microscope, we identified the sex of each gametophyte based on the presence of male or female sex expression. Each gametophyte was finely ground, and plant fragments were used to cultivate protonema of each individual. This process was repeated until many of the same, cloned individuals were growing simultaneously in the greenhouse. All plants received the same environmental conditions. The starter cultures grew in the greenhouse for 24 months before creating the experimental moss mesocosms (see below).

### (b) Mesocosm establishment and cultivation
Sixteen 38-litre pots were filled with a blend of commercial sand and peat moss (2 : 1); upon examination under a dissecting microscope, the substrate contained no discernable microarthropods. The pots provided adequate buffering from excessive cold and drought (EES and SEM 2012, unpublished data).

We added mosses to the substrate as homogenized tissue. For all 16 mesocosms, the female moss addition was the same. It consisted of the gametophytic tissue of nine female individuals, grown as described above, 4 g each, combined to a total of 36 g of female moss tissue (F1–F9). We sifted the moss tissue and homogenized the mixture in small batches using falcon tubes containing 25 ml tap water. We then combined the small moss-water batches and aliquoted the homogenized female moss-water combination into sixteen 50 ml Falcon tubes (one tube per mesocosm).

We used two different male tissue combinations, each solution consisted of three of the six males used in the experiment (M1–M6); 4 g of each male moss was combined, sifted, and homogenized in small batches, resulting in two 50 ml Falcon tubes, each containing 12 g of mixed male tissue.

We designed the mesocosms such that the male moss tissue would be applied to the centre of each pot (the mid-point of dispersal), with female mosses surrounding the males. We placed a 10.5 cm diameter plastic disc in the centre of each pot, covering the centre, while we evenly applied the female solution to the surrounding, uncovered substrate using a large wide-tipped syringe. Following this, we removed the protective disc and evenly applied one of the two male solutions to the centre of each mesocosm pot. The application density of the male moss-water extract was approximately double that that of female moss-water extract applied per area.

Moss mesocosms were initially grown in the greenhouse and were hand-misted twice daily for two months. They were kept at 18°C and a 14-hour photoperiod of approximately 200 microeinsteins (μE) and a night-time temperature of 10°C. Each mesocosm was fitted with a translucent Open Top Chamber (OTC) ring that transmits full spectrum sunlight. The OTC rings were added to serve as a barrier to prevent invertebrate immigration or emigration and to assist in providing uniform environmental conditions across all 16 mesocosms.

Two months later, the mesocosms had grown into uniform, yet still compact mats of *C. purpureus* gametophytes (less than 3 cm tall). The mesocosms were moved outside and placed on an adjacent impervious surface and except for occasional supplemental misting early in their establishment, the mosses grew in fully exposed natural outdoor conditions (Portland, OR, USA), including at least one winter freeze and snowfall event.

### (c) Microarthropod additions
To test the effects of microarthropods on moss reproduction, microarthropods were added to half (eight) of the experimental mesocosms. Microarthropods were sourced from naturally occurring mats of mosses (largely *C. purpureus*), found and collected near Portland, OR. Collected moss mats were misted with tap water, weighed into 100 g portions, and added to modified, collapsible Berlese funnels for live invertebrate extraction under 15 W incandescent bulbs [25,32–34]. A custom-designed protective canopy holding the suspended Berlese funnels was situated over the 16 mesocosms, allowing for microarthropod additions to occur without moving the experimental mesocosms. Over a year, we conducted nine 48-h live microarthropod extraction treatments. We performed one control extraction (into an ethanol solution) during each treatment allowing us to quantify, under a dissecting microscope, the average abundance and composition of invertebrate additions. There was a mean of 356 (±106 s.d.) invertebrates per addition. The extractions were largely comprised of springtails (Collembolan sp.) and mites (Oribatida and Prostigmata), as well as other invertebrates including species from: Annelida, Arachnida, Coleoptera, Diptera, Hymenoptera, and Nematoda. After the microarthropod extraction, we returned the dried source moss material to the local environment.

To assess if adding microarthropods to the mesocosms influenced moss growth (thereby increasing fitness via a mechanism other than sperm transfer), we measured chlorophyll fluorescence as the maximum quantum yield of photosystem II (PSII) (Fv/Fm) from the canopy of each mesocosm. We made measurements when sporophytes had begun to develop in 7 of 16 mesocosms (after about 12 months), and three months later. Fluorescence was measured on dark-adapted *C. purpureus* at five locations in each mesocosm before sunrise [35,36]. We also determined canopy chlorophyll content by chlorophyll fluorescence [37] (reported as CFR, chlorophyll fluorescence ratio), using a hand-held chlorophyll meter (Opti-Sciences, CCM-300 Chlorophyll Content Meter, Hudson NH, USA), using standard manufacturer recommended protocols, five values per mesocosm were averaged to obtain one data point per mesocosm.

### (d) Sporophyte collection and spore culture
After 15 months, 12 of 16 mesocosms had developed mature sporophytes, and we began collecting sporophytes. The location of each mature sporophyte was surveyed for distance from pot centre and angle vector, and carefully removed from the mesocosm with forceps—along with its maternal gametophyte when feasible. The height of each sporophyte was measured with digital calipers, recorded, and placed into a labelled conical tube.

Spores from each sampled sporophyte were grown in axenic culture for genetic analysis. The spores from the sporophytes were isolated, counted, and germinated providing a direct measure of fitness above and beyond sporophyte production [38]. In total, 325 operculate sporophytes were surface sterilized for 25 s in a 20% solution of commercial bleach (8.25% sodium hypochlorite) and triple rinsed in sterile distilled water before the spores were released into 1 ml of sterile water by mechanically

disrupting the capsule. We germinated 10 µl of spore suspension per sporophyte on BCD with ammonium tartrate (BCDA) media [39], grown at 25°C with continual light. We evaluated each inoculation of spore solution after 5–7 days to ensure there was germination from > 20 spores. After 14–21 days of growth, DNA was extracted from protonema following a modified cetyl trimethyl ammonium bromide (CTAB)/chloroform protocol [38].

### (e) Loci selection and illumina library preparation

To assess the parentage of a subset of sporophytes, we used a novel, comprehensive genotyping approach. We first chose hypervariable nuclear loci identified by McDaniel *et al.* [40]. We amplified these and Sanger sequenced loci via polymerase chain reaction (PCR) in 15 putative parents (6 males, 9 females). We then verified diagnostic single nucleotide polymorphisms (SNPs) within the putative parents using Geneious v. 8.1.8, resulting in five loci being chosen for use.

Illumina library preparation followed a modification of the Illumina 16S metagenomic protocol (Illumina no. 15044223 Rev. B) where all loci-specific primers have a 33 bp tail added to the 5′ end. This tail contains the binding site of the Illumina sequencing primers and provides a binding site for the indexes (barcodes) and flowcell binding sequences which are added in a second PCR reaction. See electronic supplementary material for more procedural details. The product of the first multiplexed PCR served as the template DNA for the second PCR where each individual's pool of PCR products was indexed with a 5′ and 3′ index. Custom indexing primers were design modelled after Illumina Nextera sequence adaptors [41]. The combination of the two indices provided a unique identifier for each individual allowing the pooling and sequencing of several hundred separate libraries in a single Illumina run. The second PCR was carried out that included 1.6 µl product from the first PCR, run for 10 cycles with a 45 s 55°C annealing temperature. PCR 2 products were visualized and cleaned, then cleaned libraries were quantified and further cleaned. MiSeq 2 × 250 bp sequencing (Illumina, San Diego, CA, USA) was performed at the University of Florida's Interdisciplinary Center for Biotechnology Research.

### (f) Genetic data processing and analysis

Raw binary base call (BCL) files, from the MiSeq, had adaptors removed, converted to fastq, and demultiplexed allowing one mismatch in the 5′ and 3′ indexes using Illumina's bcl2fastq v. 2.16.0.10 [42]. General patterns observed in FastQC quality plots were used to inform quality trimming parameters. Reads were trimmed using a 10 bp sliding window, with a minimum average quality threshold of 30 using Trim.pl [43]. Trimmed reads were then evaluated again for quality and read length distribution using FastQC. Paired end and singleton reads were assembled against the *C. purpureus* genome (v. 0.5) using Bowtie2 v. 2.2.6 [44]. Since each sample consists of a pool of progeny for each sporophyte, two haplotypes will be present (one corresponding to each parent). Each sample's binary alignment map (BAM) file was analysed using SAMtools to generate two BAM files each containing aligned sequence reads of each corresponding haplotype. SNPs found in the mpileup were called with BCFtools call v. 1.2 [45], using the multiallelic-caller (-m), ignoring indels (-V), and calling invariant sites. Genomic regions corresponding to the targeted amplicons were extracted from the variant call format (VCF) output using BCFtools filter (-r). The resulting amplicon VCFs were converted to fasta using a custom Perl script that also evaluated read depth at every position, if read depth dropped below 25 for a given position the script would return an N in the fasta sequence file, indicating the absence of sufficient sequencing data to accurately call the nucleotide at that position. Each amplicon's sequence file was combined with the Sanger sequenced data from putative parents and aligned using MAFFT implemented in

Geneious v. 8.1.8 (BioMatters Ltd, Auckland, New Zealand). The resulting alignments were clustered based on pairwise sequence similarity and every individual's two haplotype sequences were assigned to a known or unknown parent. Because each locus was not capable on its own of resolving every parent this process was repeated across all loci, ultimately producing a unique multi-locus assignment that when compared to the known parents could identify maternal and paternal contributors to the sporophyte.

### (g) Data analysis

We used a two-way ANOVA to determine the effects of microarthropod treatment, sampling date, and the interaction between these effects on CFR in the *C. purpureus* canopies, with CFR log-transformed, and ANOVA to determine the effects of microarthropod treatment on $F_v/F_m$. General linearized model (GLM) was used to test the effect of microarthropod treatment on sporophyte counts [46].

In genotyping 325 sporophytes, we found that initially planted genotypes accounted for 95.7% of the paternal genotypes and 85.8% of the maternal genotypes in our sampled sporophytes. Because in some sporophytes we did not sequence the diagnostic SNP which enabled us to distinguish between female genotypes F3 and F6, we assigned these plants to an F6′ maternal parentage. In all cases where we could distinguish between the two, the maternal parent was F3. We used $\chi^2$ tests to determine whether male genotypes differed in their success at producing sporophytes, whether female genotypes differed in their success, and whether there was variation among male genotypes in fathering sporophytes for each female genotype.

We used ANOVA to determine the effects of microarthropod treatment, paternal genotype, and maternal genotype on the distance of each sporophyte from the centre of the mesocosm [47]. For each male genotype, we calculated distribution of distances from the centre for sporophytes fathered by the genotypes, and whether the distribution was significantly similar to the normal or lognormal distribution using Kolmogorov–Smirnov tests, testing whether sperm dispersal is similar to plant propagule dispersal with a distribution with positive kurtosis (leptokurtic), and may be modelled with the lognormal distribution [48,49]. We also used ANOVA to test the effects of microarthropod treatment, paternal genotype, and maternal genotype on offspring characters including sporophyte height, per cent of spores that germinated, and number of spores produced.

## 3. Results

### (a) Microarthropods and plant parental genotype affect sporophyte production

A total of 839 sporophytes grew across the eight mesocosms with added microarthropods, significantly more than the 228 sporophytes that grew in mesocosms without added microarthropods (figure 1a; $\chi^2 = 345.64$; $p < 0.0001$; $N = 16$ mesocosms). To assess the potential effects of microarthropods on moss physiology, we measured moss chlorophyll fluorescence and chlorophyll content. The addition of microarthropods did not affect chlorophyll fluorescence ($F_v/F_m$) in two sampling periods ($F = 0.5982$; $p = 0.4458$; $N = 32$; mean ± s.e. = $0.57 ± 0.05$ and $0.58 ± 0.05$, respectively). There was no significant interaction between microarthropod treatment and sampling period on chlorophyll content (CFR, $F = 0.18$ $p = 0.68$), suggesting that the relative chlorophyll content between the treatments did not change across the experiment, nor did microarthropods discernably affect plant physiology.

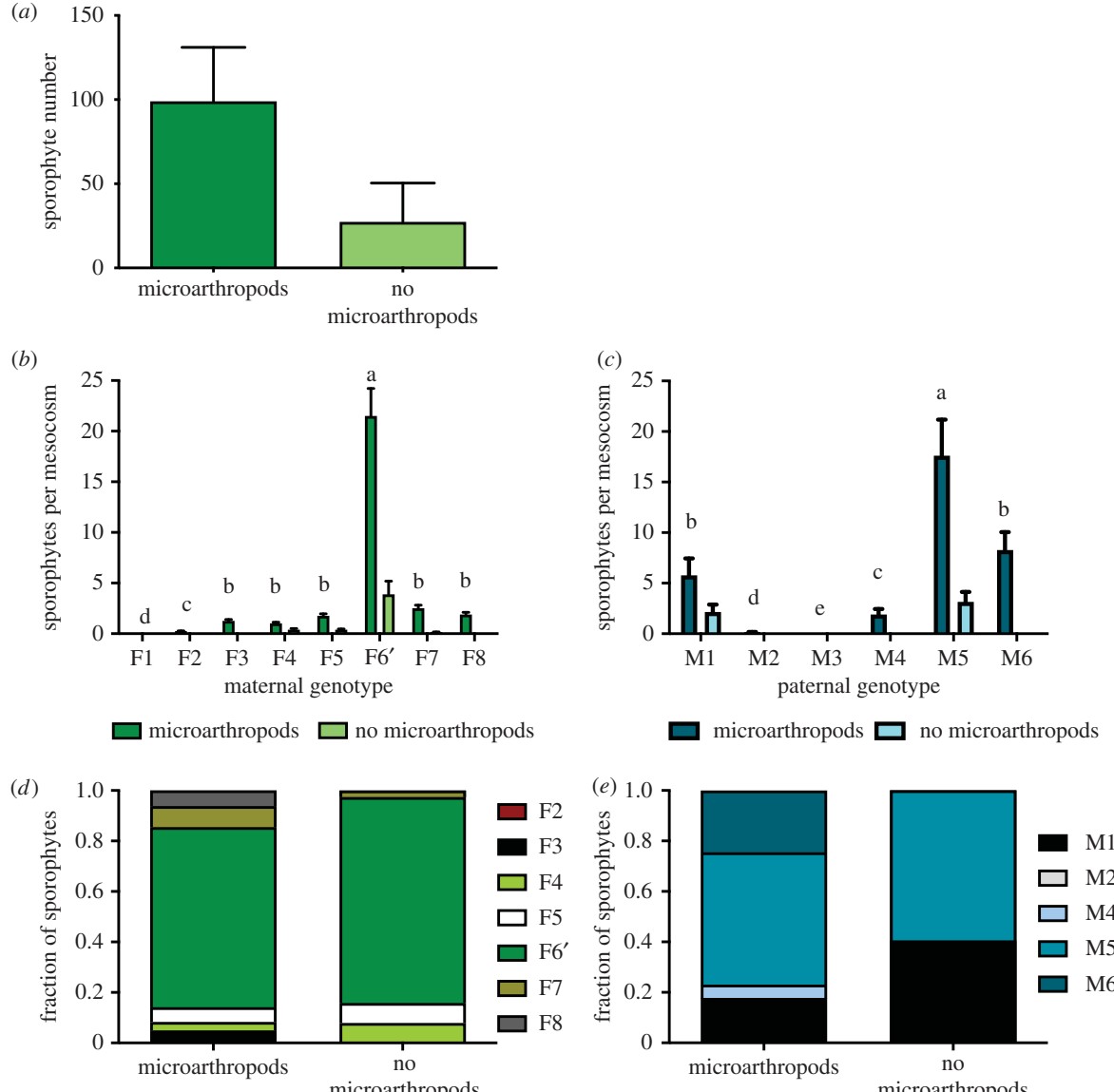

**Figure 1.** Sporophyte production in mesocosms. (*a*) Mean (+1 s.e.) sporophytes produced in mesocosms with and without microarthropods after 16 months ($N =$ 16 mesocosms with 1067 total sporophytes; $p < 0.0001$). Variation in mean (+s.e.) sporophytes per mesocosm among (*b*) female genotypes ($p < 0.0001$) and (*c*) male genotypes ($p < 0.0001$) produced in treatments with and without microarthropods. Different letters represent significant differences within genotypes for the microarthropod treatments, in which the majority of sporophytes were produced. Comparison of (*d*) maternal genotype and (*e*) paternal genotype numbers in treatments with and without microarthropods ($p = 0.07$ and $p < 0.0001$, respectively). (Online version in colour.)

The moss genotypes used in this experiment had different fitness, as measured by sporophyte production. Maternal genotype affected sporophyte number (figure 1*b* and table 1; $\chi^2 = 276.63$; d.f. $= 4$; $p < 0.0001$), with female F6′ producing the most sporophytes. F2 produced only one sporophyte, and F1 produced none. Paternal genotype also had an effect on sporophyte number, with male M5 fathering more sporophytes than the other genotypes in the population, genotypes M2 and M4 fathering fewer, and M3 fathering no sporophytes (figure 1*c*; Likelihood $\chi^2 = 284.41$; d.f. $= 4$; $p < 0.0001$). For the majority of females (F3, F5, F6′, F7, and F8), there was significant variation among males in whether they fathered sporophytes with these genotypes (table 1; d.f. $= 5$; $\chi^2 = 14.91$, $p = 0.01$; $\chi^2 = 14.91$, $p = 0.01$; $\chi^2 = 21.50$, $p = 0.0007$; $\chi^2 = 18.73$, $p = 0.002$; $\chi^2 = 14.91$, $p = 0.01$, for the five female genotypes, respectively).

Sporophyte maternal genotype was not significantly affected by microarthropod treatment ($\chi^2 = 11.80$; $p = 0.07$). However, sporophyte paternal genotype was significantly

**Table 1.** Plant parental genotype affects fitness. Boxes show the number of sporophytes produced by each potential pair of male-female genotypes. Shading reflects pairs that produced sporophytes, and darker shading reflects pairs that produced higher numbers of sporophytes.

|  | F1 | F2 | F3 | F4 | F5 | F6' | F7 | F8 |
|---|---|---|---|---|---|---|---|---|
| M1 | 0 | 0 | 0 | 4 | 2 | 47 | 5 | 0 |
| M2 | 0 | 0 | 0 | 0 | 0 | 1 | 0 | 0 |
| M3 | 0 | 0 | 0 | 0 | 0 | 0 | 0 | 0 |
| M4 | 0 | 0 | 0 | 0 | 0 | 11 | 2 | 2 |
| M5 | 0 | 0 | 10 | 6 | 12 | 89 | 10 | 7 |
| M6 | 0 | 2 | 0 | 1 | 3 | 41 | 4 | 6 |

affected by microarthropod treatment, with mesocosms with microarthropods having sporophytes fathered by five paternal genotypes while mesocosms without microarthropods had sporophytes fathered by only two genotypes (figure 1*e*; $\chi^2 = 32.08$; $p < 0.0001$).

We found that the distance of a sporophyte from the centre of the pot (farther from males) was significantly affected by microarthropod treatment ($F = 20.09$; $p < 0.0001$), and paternal genotype ($F = 14.50$; $p < 0.0001$). Females in pots without microarthropods produced more sporophytes farther from the centre of the mesocosms (farther from males) than those with microarthropods (mean ± s.e. distance from the centre $13.61 \pm 0.30$ and $11.19 \pm 0.16$ cm, respectively), although the distributions were broadly overlapping. Maternal genotype did not affect sporophyte distance from the mesocosm centre ($F = 1.33$; $p = 0.2427$).

Some male–male competition was evident in the spatial distribution of paternities from the centre of a mesocosm (dispersal distance for the sperm from the male genotypes). The M1 and M4 fertilization distance distributions did not differ significantly from lognormal distribution (Kologorov's $D = 0.059$, $p = 0.15$ and Kologorov's $D = 0.15$, $p = 0.15$, respectively, for goodness of fit to the lognormal distribution). On the other hand, the distributions of the M5 and M6 fertilization distances were significantly different from lognormal (Kologorov's $D = 0.10$, $p = 0.01$ and Kologorov's $D = 0.17$, $p = 0.01$, respectively, for goodness of fit to the lognormal distribution). The M1 fertilization distribution had a large positive kurtosis (1.39; indicating a tail away from the centre of the pot) while the distribution of M4, M5, and M6 had a negative kurtosis (−0.84, −0.90, and −1.04, respectively, indicating a tail towards the centre).

## (b) Haploid parental genotype affects diploid offspring traits

To evaluate the potential of microarthropods to influence fitness beyond sporophyte production, we estimated sporophyte height, total spore production, and average spore germination. Sporophyte height was affected by both maternal and paternal haploid genotype (d.f. = 6, $F = 11.25$, $p < 0.0001$; and d.f. = 4, $F = 4.59$, $p = 0.0014$, respectively), but not microarthropod treatment (d.f. = 1, $F = 0.22$, $p = 0.64$). Spore number was also affected by maternal and paternal haploid genotype (d.f. = 6, $F = 4.65$, $p = 0.0004$; and d.f. = 2, $F = 4.77$, $p = 0.004$, respectively; figure 2a,b), but not by microarthropod treatment (d.f. = 1, $F = 0.35$, $p = 0.55$). We found an effect of paternal genotype on spore germination rate (d.f. = 3, $F = 2.90$, $p = 0.05$; figure 2c). Maternal genotype and microarthropod treatment did not influence spore germination rates (d.f. = 5, $F = 1.57$, $p = 0.19$; and d.f. = 1, $F = 0.78$, $p = 0.38$, respectively; see electronic supplementary material, for details).

## 4. Discussion

Facilitative interactions are ubiquitous in nature, perhaps nowhere more obviously than animal dispersers of plant gametes or seeds. Animal-mediated fertilization in plants likely arose as early as the Devonian in the ancestors of modern bryophytes [21], the Triassic in gymnosperms [50], and the Cretaceous in angiosperms [51]. Nevertheless, the fitness consequences of such syndromes can be difficult to study. In angiosperms, the selection on floral features by biotic pollinators is well-examined [13–15], but the contribution of such selection to the maintenance of genetic variation is less well-understood [52,53]. Similarly, in mosses, preliminary data show that animal-mediated fertilization may influence fitness,

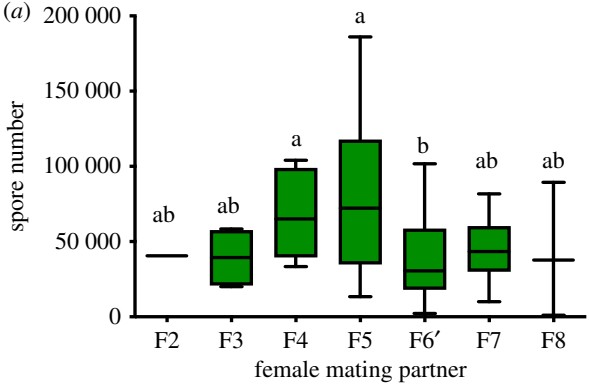

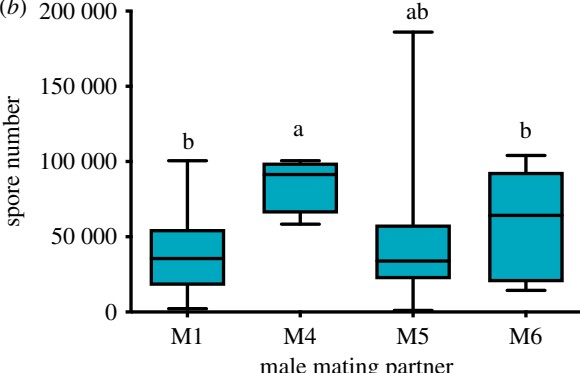

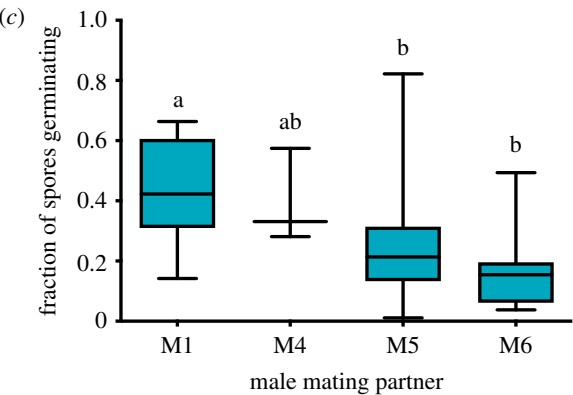

**Figure 2.** Haploid parental genotype affects diploid offspring traits. Effect of maternal and paternal genotype on offspring traits. (*a*) Spore number produced by each female mating genotype ($p = 0.0004$). (*b*) Spore number produced by each male mating genotype ($p = 0.004$). (*c*) Fraction of spores germinating by each male mating genotype ($p = 0.05$). Maternal genotype did not affect the fraction of germinating spores ($p = 0.19$). Different letters represent significant differences among genotypes. Bars represent minimum to maximum; centre line is the mean. (Online version in colour.)

but its role in the maintenance of genetic variation is unknown. Mosses produce volatile compounds that attract microarthropods [24], and in laboratory experiments, mites and springtails increase fertilization [23,24]. Here, we show that in large-scale semi-natural experimental mesocosms, the addition of microarthropods increases sporophyte formation in the moss *C. purpureus* by a factor of five (figure 1*a*). Importantly, the increase in sporophyte production is unlikely to be because adding microarthropods improves the condition of the plants, as chlorophyll fluorescence and offspring sporophyte height were equivalent between the treatment and control.

Remarkably, we found that the mosses exhibited genotype-specific responses to the addition of microarthropods. The spatial distribution of sired offspring, measured by the mean distance from the centre of the mesocosm, differed among

male genotypes. The most successful moss male genotypes M5 (the most prolific father in the treatment and controls) and M6 had a paternity skew towards the centre of the mesocosms, whereas M1, who was the second most prolific sire in the control mesocosms had the most offspring occurring furthest from the mesocosm centre. It is possible that this result is a product of timing of sperm release or another yet undetected factor. More sporophytes in the mesocosms with microarthropods developed closer to the centre of the mesocosm than those without microarthropods. Thus, the arthropods may not simply increase the fertilization distance, but they may specifically target the sperm to the archegonia, guided by sex-specific odours. It remains unclear if the microarthropods gain nutritional resources from this interaction, such as secreted sugars and fatty acids [23,54], epiphytic bacteria or fungi [55], or the moss itself [56]. These results suggest that natural populations of *C. purpureus* that lack dispersing arthropods may be sperm-limited, a result previously reported in other moss species [57].

Strikingly, in mesocosms with microarthropods, more male and female genotypes reproduced, contributing to greater genetic diversity of sporophyte offspring than in mesocosms without microarthropods (figure 1*d*,*e*). Moreover, the presence of microarthropods altered the rank order of fitness among both male and female genotypes. M6 for example, sired no sporophytes in mesocosms without microarthropods, yet had the second largest paternity contribution in the mesocosms with the microarthropods (figure 1). In females, F7 increased in rank in sporophyte production with microarthropods. In both sexes, other genotypes that did not reproduce in the control mesocosms contributed to sporophyte production when microarthropods were present. These dramatic changes in fitness suggest that the presence of microarthropods could have long-term implications for the maintenance of genetic variation within natural populations. Thus, they presumably could alter the reproductive success of moss genotypes in natural populations.

In principle, all males in a mesocosm had the opportunity to mate with all females, but surprisingly some male–female genotypic combinations were absent while others were overrepresented (table 1). Much of the variation is likely attributable to particularly competitive individuals generating more fertilizations (table 1). All males were not equally effective at fertilizing females, although whether these differences reflect male gametophyte growth or a sperm phenotype remains unknown. However, the absence of combinations of specific competitive male and female genotypes (e.g. F2, F3, or F8 with M1) suggests that some other factor, potentially related to the timing of gamete production [58], or mating compatibility, contributes to variation in sporophyte production. Another possible explanation is cryptic female choice; each female gametophyte makes several eggs, each within the archegonia, but only one ever becomes a mature sporophyte. Thus, the heterogeneous distribution of offspring genotypes in our mesocosms could result from the selective maternal support of only one fertilized egg, or an egg fertilized by a particular male sire.

We also found evidence for genetic variation for spore production among the progeny of the experimental lines used in the mesocosms (figure 2). These data suggest that the genetic variation necessary for parent–offspring conflict also is likely to be present in *C. purpureus*. In mosses, nearly all of the mineral nutrition necessary for offspring sporophyte growth is provided by the maternal gametophyte [59]. Here, we show that a single female haplotype allocated different amounts of energy, as measured by spore production, to offspring sporophytes sired by different males. Spore germination rates also varied among sporophytes, consistent with other studies in *C. purpureus* [60–62]. Researchers predicted that variation in spore production should be common, with some males extracting more nutrients from maternal gametophytes to maximize spore production [63,64] (figure 2*a*). Female traits may also influence the growth of the nutritionally dependent embryo [65,66], and indeed sporophytes sired by one male haplotype but two female haplotypes produced different numbers of spores. Consistent with these data, others found that both male and female haplotypes had a strong effect on variance of sporophyte fitness in natural populations of the moss *Sphagnum macrophyllum* [67].

Here, we show that microarthropods dramatically increased *C. purpureus* sporophyte production, increased the number of haplotypes that reproduced, and altered the rank-order fitness of haplotypes in experimental mesocosms. Even in this small sample of *C. purpureus*, individuals were highly polymorphic for previously unmeasured components of fitness related to mating, consistent with other polymorphic traits in this ubiquitous species [24,68–70]. These data demonstrate that moss mesocosms containing arrays of known genotypes, combined with pooled genotyping, can provide profound insights into the evolutionary forces shaping plant mating system biology, including sperm-limitation, male–male competition, female choice, and parent-offspring conflict, similar to such studies in angiosperm systems [70–75]. The commensal relationship between mosses and microarthropods, and its potential to influence competitive interactions among moss haplotypes, highlights the role of biotic interactions in the maintenance of genetic variation for moss fitness.

Data accessibility. Data available from the Dryad Digital Repository: https://doi.org/10.5061/dryad.crjdfn31n [76] A preprint of this article is available from bioRxiv, the preprint server for biology: https://doi.org/10.1101/2020.12.02.408872 [77].

Authors' contributions. E.E.S., S.M.E., and T.N.R. designed the experiment. E.E.S. conducted the experiment. S.F.M., A.C.P., and S.B.C. conducted the genetic analysis. E.E.S. and S.M.E. analysed the data and E.E.S., S.M.E., T.N.R., and S.F.M. wrote and revised the paper.

Competing interests. We declare we have no competing interests.

Funding. This work was supported by the National Science Foundation (DEB 1210957) to S.M.E. and E.E.S. and (IOS 128225) to T.N.R. and S.M.E., and (DEB 150041) to S.F.M.

Acknowledgements. We thank Tera Hinkley, Timea Deakova, Steven Cody Woll, Tina Arredondo, Emily Black, and Trevor Williams for their help in measuring and collecting sporophytes, as well as Linda Taylor for greenhouse support.

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
